# Impact of *Lachancea thermotolerans* on Chemical Composition and Sensory Profiles of Viognier Wines

**DOI:** 10.3390/jof8050474

**Published:** 2022-04-30

**Authors:** Ana Hranilovic, Warren Albertin, Dimitra L. Capone, Adelaide Gallo, Paul R. Grbin, Lukas Danner, Susan E. P. Bastian, Isabelle Masneuf-Pomarede, Joana Coulon, Marina Bely, Vladimir Jiranek

**Affiliations:** 1Unité de Recherche Œnologie, EA 4577, USC 1366 INRAE, ISVV, University of Bordeaux, Bordeaux INP, 33882 Villenave d’Ornon, France; warren.albertin@u-bordeaux.fr (W.A.); isabelle.masneuf@agro-bordeaux.fr (I.M.-P.); marina.bely@u-bordeaux.fr (M.B.); 2Department of Wine Science, School of Agriculture, Food and Wine, The University of Adelaide, Urrbrae, SA 5064, Australia; dimitra.capone@adelaide.edu.au (D.L.C.); adelaidegallo6@gmail.com (A.G.); paul.grbin@adelaide.edu.au (P.R.G.); lukas.danner@adelaide.edu.au (L.D.); sue.bastian@adelaide.edu.au (S.E.P.B.); 3École Nationale Supérieure de Chimie et de Physique de Bordeaux (ENSCBP), Bordeaux INP, 33600 Pessac, France; 4The Australian Research Council Training Centre for Innovative Wine Production, Urrbrae, SA 5064, Australia; 5Bordeaux Sciences Agro, 33170 Gradignan, France; 6BioLaffort, 33270 Floirac, France; joana.coulon@laffort.com

**Keywords:** *Lachancea thermotolerans*, non-*Saccharomyces* yeasts, alcoholic fermentation, wine acidification, lactic acid, wine aroma, sensory analysis

## Abstract

Viognier is a warm climate grape variety prone to loss of acidity and accumulation of excessive sugars. The yeast *Lachancea thermotolerans* can improve the stability and balance of such wines due to the partial conversion of sugars to lactic acid during alcoholic fermentation. This study compared the performance of five *L. thermotolerans* strains in co-inoculations and sequential inoculations with *Saccharomyces cerevisiae* in high sugar/pH Viognier fermentations. The results highlighted the dichotomy between the non-acidified and the bio-acidified *L. thermotolerans* treatments, with either comparable or up to 0.5 units lower pH relative to the *S. cerevisiae* control. Significant differences were detected in a range of flavour-active yeast volatile metabolites. The perceived acidity mirrored the modulations in wine pH/TA, as confirmed via “Rate-All-That-Apply” sensory analysis. Despite major variations in the volatile composition and acidity alike, the varietal aromatic expression (i.e., stone fruit aroma/flavour) remained conserved between the treatments.

## 1. Introduction

Viognier is a warm-climate grape variety traditionally associated with the world-renowned Condrieu and Château Grillet appellations in the Rhône valley [1]. It was once an obscure variety that barely covered 30 hectares in the mid-1980s, but it progressively increased in hectarage in the Old World and New World alike, from 3160 ha in 2003 to almost 12,000 ha in 2015 [1,2]. Such popularity is perhaps reflective of Viognier’s versatility as both a blending grape and in production of monovarietal wines. In blends, it commonly accompanies other warm climate whites (e.g., Rousanne, Marsanne, Maccabeo), and, in small proportions (<20%), it is also used in co-fermentations with Shiraz [3]. Monovarietal Viognier whites are full-bodied and have a distinct ‘stone fruit’ aroma, imparted by linalool, geraniol and α-terpineol [4].

Recent transcriptomic studies confirm that these monoterpenes are accumulated in Viognier berries in the later stages of ripening [5], providing support for the anecdotal evidence about longer ripening time as a pre-requisite for full flavour development in Viognier. Such practice is, however, highly conducive to accumulation of excessive sugars and loss of acidity in grapes, which are further exacerbated by accelerated phenological development in the context of a warming climate [6]. The resultant high ethanol/low acidity levels are detrimental for wine chemical and sensory profiles, microbial stability and, given the rising demand for ‘fresher’ styles, consumer acceptance and marketability. Winemakers therefore seek to address these issues through a range of external inputs and/or interventions, which can be costly, complicated and even adverse for wine quality [7,8].

Excessive ethanol levels can be decreased via different approaches that span the entire grape/wine production chain, from altered vineyard practices to partial post-fermentative dealcoholisation [7,8,9]. Recent changes in legislation in some winemaking countries (e.g., Australia, USA) allowed for the pre-fermentative water addition to dilute the initial sugar levels and thereby limit the risk of ‘stuck’ fermentation. This simple approach to lower sugar/ethanol seemingly has benign side effects on wine phenolics, aroma composition and sensory profiles [10], but, clearly it does not address the insufficient acidity in ferments. This is most commonly achieved through addition of tartaric acid, and less so with other organic acids and ion exchange techniques [11], imposing substantial costs for the winemaker. The use of an acidifying, lower-ethanol yielding yeast to conduct fermentation therefore represents a promising alternative to control both these parameters [7].

One such yeast is *Lachancea thermotolerans* (LT), a ubiquitous species associated with a range of ecological niches worldwide [12]. It commonly occurs in wine-related environments, and has thus been explored for its oenological application [13,14]. As a result, several LT starters are nowadays commercially available to be used in co-cultures with either simultaneously or sequentially inoculated *Saccharomyces cerevisiae* (SC), required to ferment to ‘dryness’ [15].

The main metabolic contribution of LT is L-lactic acid biosynthesis from sugars during alcoholic fermentation. This occurs via lactic acid dehydrogenase (LDH) activity from pyruvate obtained though glycolysis (i.e., breakdown of sugars), and is thus a pathway competing with ethanol production. The highest reported concentration of lactic acid formed by LT under oenological conditions exceeds 16 g/L [16], which is unique among non-genetically modified yeasts [17]. By comparison, SC strains produce very little, if any, lactic acid [17]. Lactic acid production in LT is a highly strain-dependent trait [16,18]. For example, concentrations of lactic acid formed in fermentations in the same grape juice using 94 different strains ranged between 1.8 to 12 g/L, and significantly affected the wine pH (3.2–3.8) [18].

In mixed cultures of LT and SC, lactic acid levels depend on the LT strain but also on the yeast inoculation regime. Due to antagonism by SC of LT, mediated by mechanisms of cell–cell contact and secretion of antimicrobial peptides [19], co-inoculation generally yields less lactic acid compared to sequential inoculation [20,21,22]. The extent of wine acidification in LT wines is thus variable, from comparable to about 0.5 units lower in pH relative to the SC monocultures [20,22,23]. Accordingly, wines co-fermented with LT contained either similar or about 1% *v*/*v* lower ethanol concentrations than the SC controls [20,22,23,24,25,26]. Other compositional modulations in LT wines include increases in glycerol [20,21,22], decreases in acetic acid [21,24,27], partial degradation of malic acid [18,20,26,28] and alterations of a range of both grape- and yeast-derived aroma compounds [18,22,25,26,28,29]. The effects of LT modalities on sensory perception of the wines have also been reported [20,22,23,26,27], as well as their impact on malolactic fermentation [30].

Our previous work focused on the population-wide study of genetic [12] and phenotypic [18] diversity in LT, which helped guide the selection of superior starters for wine acidity and ethanol modulation. Upon the characterisation of LT pure cultures, five genetically and phenotypically divergent strains (i.e., LT1–LT5) were tested in both co-inoculations and sequential inoculations with SC, and compared to the SC and un-inoculated controls in Merlot [26]. Depending on the strain and inoculation, pH and ethanol levels in mixed-culture dry wines were either comparable, or significantly lower than in controls (decrease of up to 0.5 units and 0.90 % *v*/*v*, respectively) [26]. Here, this unique characterisation of strains representing the diversity within the entire species was extended to the white variety, Viognier, under dramatically altered oenological conditions, such as the fermentation matrix, temperature and management. This delivered detailed information on the performance of tested treatments, in terms of fermentation and acidification dynamics, and comprehensive chemical and sensory profiles of the wines, whilst highlighting the robustness of certain LT modalities in production of ‘fresher’ wines without relying on external inputs.

## 2. Materials and Methods

### 2.1. Grapes and Winemaking

Viognier grapes were handpicked from the experimental Coombe vineyard (Waite Campus, University of Adelaide, SA) on the 21 February 2019 at an estimated 13.5 Bé. The grapes were stored at 0 °C prior to destemming/crushing (Bucher Vaslin Delta E2, Chalonnes-sur-Loire, France) and pressing (Bucher Vaslin XPro8 pneumatic press, Chalonnes-sur-Loire, France). Approximately 200 L of juice was collected at the press, with the addition of potassium metabisulfite (PMS; 120 mg/L) to yield approximately 60 mg/L of total SO_2_. After settling (2 days at 0 °C) and racking, 3 L of juice was aliquoted into each fermenter (5 L glass demijohns). The initial total soluble solids (TSS) were 13.5 Bé (24.4 °Brix), pH 3.9 and yeast assimilable nitrogen (YAN) 140 mg/L (106 and 34 mg/L amino acids and ammonia, respectively). The fermenters were transferred into a temperature-controlled room (~17 °C), and once acclimatised, they were inoculated as described below. After 24 h, 100 mg/L of YAN as diammonium phosphate (DAP, 10% aqueous solution) was added to each fermentation. The TSS and pH were monitored regularly throughout fermentation using a digital density meter (DMA 35, Anton Paar, Graz, Austria) and a pH meter (PH222 Lutron Electronic, Taipei, Taiwan), respectively. After TSS dropped below 0 °Bé, residual sugars were tested spectrophotometrically (Infinite 200 PRO, Tecan, Männedorf, Switzerland) using an enzymatic kit (K-FRUGL, Megazyme, Ireland) in a 96-well plate format. Upon completion, wines were stabilised and conserved at 0 °C until bottling (0.75 L; crown-seal) with the addition of 30 mg/L PMS. Carbon dioxide (as dry ice) was used at all stages of winemaking to minimise oxidation.

### 2.2. Yeast Treatments

Tested yeast treatments included five LT strains in two inoculation modalities, and a monoculture of an SC strain (Zymaflore^®^ Spark, Laffort, France), in triplicate. An un-replicated un-inoculated treatment was also included, to ascertain that the observed differences were attributable to the inoculated yeast(s) rather than the indigenous ones on the grapes/winemaking equipment.

The LT strains represented three commercially available starters (LT3, LT4, LT5) and their two experimental counterparts (LT1 and LT2). The commercial strains were sourced from various manufacturers (i.e., AEB, Italy; CHR Hansen, Denmark; Lallemand, Canada), while the LT1 and LT2 (also known as ISVV Ltyq 25 and UNIFG 18, respectively) were pre-selected as superior wine starters [12,18]. In co-inoculations, denoted with the symbol ‘x’ (e.g., LT1xSC), LT and SC strains were simultaneously inoculated at 3 × 10^6^ and 1 × 10^6^ cells/mL, respectively. In sequential inoculations, denoted with the symbol ‘…’ (e.g., LT1…SC), LT strains were added at 2 × 10^6^ cells/mL, followed 48 h later by the inoculation of SC at 1 × 10^6^ cells/mL. The SC-only treatment was inoculated at 2 × 10^6^ cells/mL, whereas any inoculation was omitted in the un-inoculated (UN) treatment. All inoculated strains were grown from cryo-cultures (−80 °C in 25% glycerol) on YPD plates (1% yeast extract, 2% peptone, 2% glucose and 2% agar) at 24 °C. After 3 days of incubation, single colonies were transferred into YPD broth (50 mL in 200 mL flasks) for an overnight incubation at 24 °C. The filter-sterilised diluted grape juice (45% water, 5% YPD; 300 mL in 800 mL flasks) was then inoculated at 10^7^ cells/mL, and incubated overnight (24 °C, 120 rpm) to permit the final inoculation rates reported above. Cell densities of the liquid cultures were determined via flow cytometry (Guava easyCyte 12HT, Merck, Kenilworth, NJ, USA).

### 2.3. Chemical Analysis

Wine ethanol concentrations were determined with an alcolyser (Anton Paar, Graz, Austria), and pH and titratable acidity (TA) with a pH meter (CyberScan 1100, Eutech Instruments, Thermo Fischer Scientific, Waltham, MA, USA) and an autotitrator (Mettler Toledo T50, Columbus, OH, USA), respectively. High performance liquid chromatography (HPLC) was used to measure the concentrations of glycerol, lactic and malic acid. Before injection (20 µL), samples were pre-filtered (syringe filter; 0.45 µm) and diluted in deionised water (2:1; final volume 2 mL). The Agilent 1100 instrument (Agilent Technologies, Santa Clara, CA, USA) was fitted with a HPX-87H column (300 mm × 7.8 mm; BioRad, Hercules, CA, USA). The eluent was 2.5 mM H_2_SO_4_, with a 0.5 mL/min flow rate at 60 °C for a 35 min run time. Signals were detected using an Agilent G1315B diode array detector and G1362A refractive index detector. Analytes were quantified using external calibration curves (R^2^ > 0.99) in ChemStation software (version B.01.03). Acetaldehyde, pyruvic and succinic acid were measured using the respective enzymatic kits in a 96-well plate format (K-PYRUV, K-ACHYD, K-SUCC, Megazyme, Ireland). Concentrations of total SO₂ were measured using an aspiration/titration method [31]. The analysis of volatile compounds was performed as described in Wang et al. [32]. The wine sample (0.5 mL) was transferred to a solid phase microextraction (SPME) vial (20 mL; screw cap vial), diluted with deionised water (4.5 mL) upon addition of sodium chloride (2 gm). The samples were spiked with a mixture of six deuterium-labelled standards (i.e., d4-3-methyl-1-butanol, d3-hexyl acetate, d13-1-hexanol, d5-ethyl nonanoate, d5-2-phenylethanol and d19-decanoic acid) as per Wang and coworkers [32].The samples were stored at 4 °C until analysis with a Gerstel MPS auto sampler (Lasersan Australasia Pty Ltd. Robina, QLD, Australia) utilising head space SPME (HS-SPME) injection, with a DVB/CAR/PDMS fibre (50/30 µm, 1 cm, 23 gauage; Supelco, Bellefonte, PA, USA). This was injected on an Agilent 7890A gas chromatograph (GC) combined with a 5975C inert XL Mass Spectrometer (MS; Agilent Technologies, Santa Clara, CA, USA), with conditions detailed in Wang, Capone, Wilkinson and Jeffery [32].

### 2.4. Sensory Analysis

The expert panel first tasted the wines in order to assure the absence of faults and consistency between treatment replicates, and defined a list of attributes to be used in the formal sensory evaluation using Rate-All-That-Apply (RATA) methodology. RATA is a rapid sensory profiling method in which the assessors are presented with a list of attributes and instructed to rate the intensity of only those that they perceive in the samples [33]. Previous research showed that RATA profiles are comparable to those obtained by the costlier and lengthier descriptive analysis [33].

Experienced wine tasters (n = 48, average age 34 years) were recruited among the post-graduate students and staff in the Department of Wine Science at the University of Adelaide. Wines were equilibrated to room temperature (22–24 °C) before pouring, and the triplicates of each treatment were blended together as they were not perceived as being different by the expert panel. Wine samples (25 mL) were presented in opaque ISO-standard glasses, labeled with four-digit-codes, and covered with glass Petri dishes. Wines were served sequentially and monadically in a random order, with 1-minute break enforced between samples, during which assessors cleansed their palates with crackers and water to overcome carry over effects.

The assessors were instructed to use a seven-point scale (1 = extremely low, 4 = moderate intensity, 7 = extremely high) to rate the applicable aroma attributes (orthonasally), flavour attributes (retronasally), and attributes related to taste, mouthfeel and length upon expectoration. In addition, the assessors were asked to indicate which attribute best described the wine acidity profile: ‘flat/flabby’, ‘bright/crisp’, ‘sour/tart’ or ‘harsh/acrid’. Wines were evaluated in individual booths at the University of Adelaide, Waite campus, at room temperature, and the data were collected using RedJade online software (Redwood City, CA, USA).

### 2.5. Statistical Analysis

Data were analysed with in-house scripts in R (version 4.0) [34]. Fermentation and acidification dynamics were analysed using k-means clustering to resolve different profiles of similar fermentation and acidification kinetics (*cutRepeatedKmeans* function; *ClassDiscovery* package). The chemical parameters of wines produced with the 11 yeast treatments were subject to one-way ANOVA, followed by Tukey’s post-hoc comparisons (*agricolae* package). Because of lack of replication, UN treatment was excluded from the statistical analysis, but retained for the graphical representation of the results. The subset of 10 LT wines was then subjected to two-way ANOVA to examine the effect of five LT strains in two inoculation modalities. The sensory data were analysed using a two-way ANOVA with panellists as random and samples as fixed factors. The significance thresholds for all ANOVA were set at 5%, and *p*-values were corrected for multiple tests (Benjamini-Hochberg correction). Principal component analysis (PCA) was used as a multivariate analysis to visualise the factors explaining most of the variation of the whole dataset and identify the possible correlations of the chemical data set.

## 3. Results

### 3.1. Fermentation and Acidification Kinetics

K-means clustering resolved three different fermentation kinetics profiles, and four acidification kinetics profiles (Figure 1). The fastest fermentation (~20 days) was displayed by Profile 1′, comprised of the SC control, all LT co-inoculations and sequential inoculations with LT3 and LT4. Comparatively slower fermentation Profile 2′ contained sequential inoculations with LT1, LT2 and LT5, which took approximately four days longer (~24 days) to complete compared to Profile 1′. Fermentation Profile 3′ corresponded to the UN treatment, which showed a lengthy lag phase of approximately seven days, then a rate similar to Profile 2′ combining to yield an overall fermentation duration of ~30 days.

The trends in pH showed initial drops at the onset of fermentation, followed by increases from day four onwards (Figure 1). The pH levels of the k-means Profile 1 (comprised of SC, UN, LT4 co-inoculation and both LT3 treatments) remained the highest, followed by Profile 2 (with LT1, LT2 and LT5 co-inoculations and LT4 sequential inoculation). The acidification was more pronounced in sequential inoculations of LT1 and LT5 (Profile 3) and LT2 (Profile 4), with the latter treatment showing the largest drop in pH of approximately 0.5 units (Figure 1). The acidifying LT treatments thus displayed slower fermentation rate than the SC control and the non-acidifying LT treatments. The impaired fermentation rate of the UN treatment was not linked to acidification but rather to the omission of inoculation.

### 3.2. Main Oenological Parameters of the Viognier Wines

The concentrations of residual sugars (RS) ranged between 1.5 (LT5xSC) and 4.2 g/L (UN), thus the wines were borderline dry. The main residual hexose in all treatments was fructose (Table 1). With a range between 14.45 and 14.90% *v*/*v* in LT2…SC and LT3…SC, respectively, the differences in wine ethanol content were significant (*p* = 0.0028), albeit marginal. The SC control had comparable ethanol levels to all other wines (14.76% *v*/*v*). Yeast treatments had a more pronounced effect on the wine pH and TA (Table 1), which were in line with the acidification Profiles (Figure 1). The sequential inoculations with LT2 had the lowest wine pH (3.62), followed by LT1 and LT5 (3.83 and 3.89, respectively). Despite slightly higher values, pH values of wines in acidification Profile 1 (SC, UN, LT4xSC and both LT3) were statistically comparable to those in Profile 2 (co-inoculations of LT1, LT2, LT5 and LT4…SC). Corresponding trends were confirmed for the wine TA values, which were around 4.6 g/L for the non-acidified wines (Table 1). In LT2…SC, TA increased to a remarkable 8.8 g/L, induced by the production of 5.2 g/L of lactic acid (Table 1). The TA and lactate levels of the remaining two bio-acidified wines (LT1…SC and LT5…SC) were intermediate between LT2 and the other treatments. The wines differed marginally (up to 0.4 g/L) in glycerol concentrations, which were slightly higher in sequential inoculations than in co-inoculations, but altogether similar to the SC control (5.62 g/L). Four LT treatments had comparable and six had lower acetic acid than the SC control (0.46 g/L). Malic acid levels in the SC control (2.97 g/L) and LT co-inoculations were higher than in all co-inoculations except LT3. The LT2…SC wine had the lowest malic acid levels, i.e., 0.4 g/L less than the control (Table 1). The same wine had the lowest concentrations of succinic acid (1.7 g/L), which in the remaining wines exceeded 2 g/L (Table 1). Significant effects of yeast treatment (>80% variation; *p* < 0.0001) were also recorded in acetaldehyde and pyruvic acid concentrations, which ranged between 26 and 77 mg/L, and 39 and 73 mg/L, respectively (Table 1). Total SO_2_ analysis revealed surprisingly high levels of SO_2_ in certain treatments, notably 102 mg/L in SC and LT3…SC (Table 1). As the grape juice was supplemented with 60 mg/L of SO_2_ during processing, this indicated production of a minimum of 40 mg/L of SO_2_ during fermentation. The co-inoculated LT treatments contained total SO_2_ levels that were comparable to the SC control. Conversely, all sequential inoculations, except LT3, had comparatively lower SO_2_ levels, which were at their lowest in the UN wine (Table 1).

The 2-way ANOVA of 10 LT treatments further confirmed that all basic oenological parameters except glycerol were more affected by the LT strain than the inoculation modality (Figure 2, Appendix A). Moreover, the inoculation modality and the strain–inoculation interaction had similar effects on the lactic acid production and the resultant pH/TA modulations. The inoculation modality was not a significant factor for acetaldehyde and acetic acid formation, unlike its interaction with the strain, suggesting that certain combinations can lead to increases in these potentially detrimental compounds (Figure 2, Appendix A).

### 3.3. Volatile Composition of Viognier Wines

A total of 29 volatile compounds were quantified in Viognier wines, including 10 ethyl esters, 3 acetate esters, 8 higher alcohols, 5 acids and 3 terpenes (Table 2 and Appendix A). Significant yeast treatment-derived differences (ANOVA α = 5%; Tukey’s post-hoc) were detected in all compounds except the varietal compound linalool (Table 2). Besides their concentrations, these were also considered in terms of their odour active values (OAV), which, despite perceptive interactions, serve as indicators for the contribution of each compound to aroma perception. This revealed that, in all wines, 14 compounds were detected above, and 12 below, their respective aroma detection thresholds, while 3 compounds (i.e., ethyl 2-methylpropanoate [sweet aroma], ethyl decanoate [floral/fruit aroma] and 2-phenylethanol [rose aroma]) exceeded their thresholds in some wines, but not others (Table 2).

Ethyl acetate was the quantitatively predominant ester in all wines except LT2…SC, which instead contained more ethyl lactate. The detected levels of ethyl acetate were relatively low, and well below the point where it would be seen as a fault rather than ‘fruity’/complexing’ (i.e., 150 mg/L) [11]. Apart from an increase in the LT5…SC treatment (66 mg/L), the LT modalities had similar amounts of ethyl acetate as the SC control (46 mg/L; Table 2). The concentrations of ethyl lactate (buttery/coconut aroma) ranged between 0.5 and 79 mg/L in SC and LT2…SC, respectively, and thus displayed the largest variation among all the analytes (Table 2). As a result of ethyl lactate increases, LT2…SC contained the highest levels of total ethyl esters (Figure 3). However, as ethyl lactate concentrations remained below the sensory threshold (i.e., 146 mg/L) in all treatments including LT2…SC (Appendix A), it is unlikely that such increases enhanced the aroma of this wine.

Albeit present at lower concentrations than ethyl acetate/lactate, ethyl butanoate, ethyl hexanoate and ethyl octanoate had comparatively higher OAV values with fruity aroma qualities (Appendix A). Irrespective of the inoculation modality, these compounds were higher in LT3 and LT4 treatments than in LT1, LT2 and LT5, as were ethyl decanoate and ethyl 2-butenoate (Table 2). In fact, LT strain explained more than 50% of variation in these compounds (Appendix A). In SC control, ethyl esters of straight-medium chain fatty acids (MCFA) were increased to equivalent levels as in LT3/LT4, suggesting that certain LT modalities are linked to their decrease compared to the controls (Table 2).

The respective MCFA precursors (i.e., butanoic, hexanoic, octanoic and decanoic acid) followed trends corresponding to those of their ethyl esters, and were higher in SC, LT3 and LT4 than in LT1, LT2 and LT5 wines. Besides significant LT strain effects, inoculation regimes also had a significant impact on their concentrations, with increased values in co-inoculations than in sequential inoculations (Figure 2, Table 2 and Appendix A). As a result, the total concentrations of fatty acids were higher in SC, both LT3 and LT4 modalities, and co-inoculations with LT1 and LT2 than in the remaining treatments (Figure 3).

Of the determined acetate esters, only isoamyl acetate (banana aroma) was above the sensory threshold in all wines. It was present in the highest concentration in LT5…SC, which was also abundant in hexyl acetate (sweet aroma) and 2-phenylethyl acetate (rose/honey aroma) (Table 2). Accordingly, the highest levels of total acetate esters were present in LT5…SC potentially due to increased activities of acetyltransferases. The lowest levels of total acetate esters were in both LT1 treatments and LT2 co-inoculation, while those in SC were intermediary (Figure 3). The LT strains explained most variation in individual and cumulative acetate esters (>64%), and their interaction with the inoculation modality was also significant (>10%).

The most prevalent higher alcohol in all wines (59% of total higher alcohols) was 3-methyl-1-butanol (isoamyl alcohol), followed by propanol (Table 2). Both of these higher alcohols were present in the LT modalities in similar amounts as in the SC control (Table 2). In LT wines, interaction between the strains and inoculation regimes significantly explained the variation in 3-methyl-1-butanol, and, consequently, total higher alcohols (Figure 3). Conversely, LT strain accounted for 54% of the variation in propanol, with superior amounts produced by LT3 (Figure 2). Despite being present at lowest concentrations, octanol had the second highest OAV values among higher alcohols, surpassed only by propanol (Appendix A). Octanol was highest in the SC control, intermediary in LT co-inoculations and further decreased in sequential inoculations (Figure 2, Table 2). Inoculation regime was thus the main explanatory factor for octanol (Figure 2, Appendix A). The concentrations of 2-phenylethanol (rose aroma) were below its sensory threshold (14 µg/L; Appendix A) in SC and four LT wines, and above it in UN and six LT wines (Table 2 and Appendix A). The LT strains accounted for 52% of variation in 2-phenylethanol, which was higher in LT1, LT2 and LT5 than in LT3 and LT4 (Figure 2, Appendix A). Butanol, isobutanol, hexanol and benzyl alcohol were lower than their respective sensory thresholds in all wines (Table 2). This was also the case for limonene and α-terpineol, which were present in higher amounts in sequential inoculations of LT2 and LT5, while linalool and total terpenes remained unaffected by the yeast treatments (Table 2).

### 3.4. Multivariate Analysis of Wine Chemical Parameters

Besides the univariate analysis, the entire chemical dataset comprised 43 parameters, which included both oenological parameters and volatile compounds, was also subjected to PCA. The first two principal components, PC1 and PC2, accounted for only 41.71% and 16.64% of the explained variance, respectively (Figure 4). Sequential inoculations of LT1, LT2 and LT5 were separated from the SC control and the remaining wines on PC1 (Figure 4). The separation of these acidifying LT treatments was driven primarily by increases in lactic acid and, consequently, TA and ethyl lactate, as well as the terpenes (limonene and α-terpineol). With the exception of further-diverged LT5…SC, the co-inoculations grouped closer to the SC control, as did the sequential inoculations of LT3 and LT4. This was driven by higher wine pH, increased abundance in MCFA and their ethyl esters, and certain organic acids (i.e., acetic, succinic and pyruvic) and higher alcohols (i.e., octanol and benzyl alcohol). The separation on PC2 was driven by increases in ethyl acetate and isoamyl acetate, resolving both LT1 treatments and, to a degree, LT2…SC, from LT5…SC, LT4xSC, both LT3 wines as well as UN. The separation of wines altogether highlighted their distinct chemical profiles resulting from the use of different yeast treatments.

### 3.5. Sensory Profiling

Comprehensive sensory profiling of wines on 34 attributes was undertaken by a large cohort of wine experts (n = 48) using RATA methodology (Appendix A). Significant differences were not detected in the aroma, i.e., orthonasal perception, of wines (Appendix A). The highest-scored aroma descriptor across all wines was for ‘stone fruit’ (mean score 3.4), followed by ‘apple/pear’ and ‘tropical’ (mean scores 3.3 and 3.1, respectively). Importantly, the fault-related attributes in all wines remained extremely/very low, with the average scores of 1.2 for ‘VA’ and ‘medicinal/rubbery’, and 1.7 in ‘oxidation’ (Appendix A). The flavour of wines, i.e., retronasal perception, corresponded to their aroma, with ‘stone fruit’ as the main characteristic (mean score 3.1).

However, despite their similar aroma profiles, the wines significantly differed in two flavour attributes, i.e., ‘citrus_F’ and ‘banana_F’ (Figure 5). The LT2…SC wine scored the highest in ‘citrus_F’ (mean score 4.3) and the lowest in ‘banana_F’ (mean score 1.4). In line with the decreased pH/increased TA (Table 1), the same wine scored the highest in ‘acidity’ (5.2), followed by the LT5…SC (4.5), while the non-acidified wines had up to 1.6 higher ‘acidity’ scores (Figure 5). The corresponding trends were detected for wine ‘acidity length’ and opposite for ‘sweetness’, with the wines with lower acidity scored as sweeter (Figure 5). With a range between 3.3 and 4.1 in LT2…SC and LT1…SC, respectively, the differences in ‘balance’ between all the taste/mouthfeel attributes were significant, albeit low (Figure 5). The ‘balance’ of the SC control wine was intermediary (3.6).

Further insight into the acidity perception was obtained by analysing the distribution of responses related to the attributes which best described the acidity profiles of the wines (Figure 5, Appendix A). The majority of panellists described the SC control as ‘flat/flabby’, which was also the main acidity descriptor for both LT3 treatments, co-inoculations with LT1 and LT4 and UN wine. The acidity profiles of co-inoculations with LT2 and LT5, and sequential inoculation with LT4, were slightly shifted towards the ‘fresh/crisp/bright’ profile, which was even more pronounced in LT1…SC and LT5…SC wines. The acidity profile of the LT2…SC wine differed from the remaining ones, as it was perceived as ‘sour/tart’ by 60% the panellists (Figure 5, Appendix A).

## 4. Discussion

Viognier is a warmer climate grape variety that often benefits from bio-acidification and a decrease in ethanol content; both were shown through this study to be attainable with the use of LT co-starters. In this study, fermentation and acidification performance of five LT strains in two inoculation regimes (i.e., co-inoculation and sequential inoculation) were compared to that of the SC monoculture, and an un-replicated treatment without inoculation was included as it further suggested that the observed wine modulations can be attributable to the inoculated yeasts rather than any other microorganisms present on the grapes.

Relative to the SC control, efficient acidification occurred in three LT treatments, i.e., sequential inoculations of LT1, LT2 and LT5, while all co-inoculations and sequential inoculations with LT3 and LT4 remained non-acidified (Figure 1, Table 1). A link between fermentation kinetics and acidification was apparent, with delayed fermentation completion in the bio-acidifying treatments (Figure 1), likely explained by microbial competition and the inhibitory effect of lactic acid [35]. The bio-acidified treatments resulted in the most divergent wine profiles as confirmed by PCA of the chemical data set (Figure 4) and RATA sensory profiling (Figure 5). The co-inoculations thus had minor impact on Viognier wine profiles. Lower metabolic contribution of co-inoculated as compared to sequentially inoculated LT modalities has been widely recognised [20,21,22], and is generally attributed to the antagonistic effects of SC upon LT [19]. However, identical LT co-inoculations had stronger impact in Merlot than in Viognier, likely due to more favourable LT growth conditions in the former matrix, in particular higher temperatures [26]. In sequential inoculations, less modulation of wine profiles by LT3 and LT4 as compared to the other three strains could potentially be attributed to their lower implantation upon inoculation. However, the performance of LT strains largely corresponded to that recorded in Merlot [26], as well as in their pure cultures [18]. Of particular interest is the contrasting behaviour of LT2 and LT3 strains, representatives of two genetically differentiated subpopulations, i.e., ‘Domestic 2′ and ‘Domestic 1′ [12], characterised as acidifying and non-acidifying strains, respectively [18]. The genetic mechanisms of differential lactic acid formation remain unclear. Only recently, Sgouros et al. [25] reported an implication of *LDH2*, but not of other *LDHs* or alcohol dehydrogenase genes, in increased lactate production. Sequential inoculation with LT2 resulted in the strongest acidification, i.e., a decrease in pH of up to 0.5 units, and an increase in TA of about 4 g/L, relative to the LT1 among the other non-acidified treatments, triggered by production of ~5 g/L of lactic acid (Table 1).

In accordance with partial diversion of sugars from ethanol to lactic acid, LT2….SC wine had the lowest ethanol concentration, with the decrease of ≤0.4% *v*/*v* relative to the non-acidified treatments. However, the differences between the SC and LT treatments were not significant and the decrease in ethanol was considerably lower than the one observed with sequential inoculation of the same strain in Merlot (i.e., 0.9% *v*/*v*) [26]. By comparison, the largest ethanol decrease to date reported in dry LT wines is 1.6% *v*/*v*, which was achieved in sterile fermentations sequentially inoculated with SC at 1% *v*/*v* ethanol [20]. However, in non-sterile conditions, the same strain and inoculation regime resulted in an ethanol drop of only 0.3% *v*/*v* [20], together showing that the fermentation matrix and conditions play an important role in modulations of ethanol content by different LT strains.

The bio-acidified LT wines were also associated with higher levels of glycerol, and lower levels of acetic acid (Figure 4). After ethanol, glycerol is generally the second most abundant metabolite formed by fermenting yeasts and a common carbon sink in lower-ethanol wines [36]. However, pure LT cultures do not necessarily produce more glycerol than SC [22,37]; increases in glycerol thus seemingly occur in response to LT-SC co-culturing. In SC, glycerol is formed via glycerol 3-phosphate dehydrogenases (GPD) to eliminate excess cytosolic NADH under anaerobic conditions, and the expression of *GPD1* and *GPD2* homologs is induced by osmotic stress and anoxia, respectively [38]. Initial glycerol formation by SC is coupled with acetic acid production to restore the redox balance [38]. Here, links between the two metabolites remained undetected, and the LT wines that were high in glycerol contained either lower or comparable acetic acid levels as the SC control (Table 1, Figure 4). The wines also differed in levels of malic, succinic and pyruvic acid, which were all lower in bio-acidified treatments (Figure 4). Decreases in malic acid in these treatments agree with previously reported partial degradation of malate in pure LT cultures [18] and co-cultures [20,28] alike. Of further interest were higher total SO_2_ concentrations in SC wines (Table 1). These trends agreed with previous reports [25,27] and warrant further investigation and application in production of wines with lower SO_2_ content.

The analysis of volatile compounds revealed marked differences in volatile composition of the wines. The analytes predominantly represented major yeast-derived metabolites, alongside several grape-derived compounds of importance for varietal Viognier aroma (i.e., terpenes, Table 2). The largest variation (0.5–79 mg/L) was detected in ethyl lactate concentrations, which in LT2…SC wine quantitatively surpassed ethyl acetate (Table 2). Ethyl acetate is generally the most abundant ester formed during AF, while the concentrations of ethyl lactate generally increase upon MLF [11]. Certain LT modalities, however, are conducive to accumulation of this ester because of the availability of lactic acid as its precursor. Accordingly, ethyl lactate levels were linked to variation in lactic acid (0.1–5.2 g/L) with increases in the bio-acidified wines, but, unlike in Merlot [26], they remained below its relatively high sensory threshold (i.e., 146 mg/L; Appendix A).

Conversely, 2-phenylethanol and ethyl 2-methylpropanoate (imparting rose and fruity aroma, respectively) surpassed their aroma detection limits in some bio-acidified wines (Table 2, Figure 4). Increases in 2-phenylethanol were previously observed in mixed fermentations with LT strains, but not necessarily in their monocultures [22,23,24], potentially due to its role as a signalling molecule [39]. Higher production of ethyl 2-methylpropanoate in LT sequential cultures agrees with several previous studies [20,28], including our trial carried out in Merlot [26]. Another corresponding trend between our two studies was the inferior levels of MCFA and their ethyl esters in sequential inoculations as compared to the SC monoculture (Figure 3 and Figure 4). The precursors of these ethyl esters (i.e., isobutyric acid of ethyl 2-methylpropanoate and MCFA of their ethyl esters) are formed via different metabolic pathways. As a branched-chain fatty acid, isobutyric acid is produced from valine via the Ehrlich pathway, unlike the MCFA, that are formed from acetyl-CoA through the fatty acid synthase (FAS) complex [11]. These observations therefore invite further research to investigate the differences between LT and SC in amino acid metabolism and biosynthesis of fatty acids and/or the release of medium-chain intermediates available for esterification, and their modulation in response to co-culturing. Further research is also required to understand yeast-derived differences in acetate esters, as in contrast to ethyl esters, their concentrations depend more on the enzymatic activities, in particular acetyltransferases, than substrate availability [11]. Different activities of β-glucosidase could account for the variation in predominantly grape-derived terpenes, i.e., over-production of limonene and α-terpineol in bio-acidified wines (Table 1, Figure 4).

Despite significant differences in volatile composition of wines, their aromatic perception remained unaltered, with a strongly expressed ‘stone fruit’ character (Appendix A) that is typical for Viognier [4]. In line with the main oenological parameters (Table 1) and our previous results in Merlot [26], acidity was not only the most discriminative sensory attribute, but also the main driver of differences in other sensory attributes (Appendix A). For example, despite comparable RS levels (Table 1), the bio-acidified wines scored lower in ‘sweetness’, and higher in ‘citrus’ flavour, highlighting the role of acidity in modulating the perception of sensory parameters other than sour taste. Such a phenomenon of combining information from multiple sensory modalities is referred to as synesthesia [11].

## 5. Conclusions

In conclusion, this work delivers chemical and sensory profiles of Viognier wines produced in co-inoculations and sequential inoculations with five LT strains, and the SC monoculture and un-inoculated controls. The modulation of the analysed wine parameters depended on both LT strains and inoculation modalities, with comparable, albeit less pronounced, trends to those observed in our Merlot winemaking trial [26]. In particular, our results highlighted the dichotomy between the bio-acidified and non-acidified LT wines, and among the former, superior performance of LT2…SC treatment to boost the acidity and marginally decrease the ethanol content without affecting the varietal typicity of Viognier wines.

## Figures and Tables

**Figure 1 jof-08-00474-f001:**
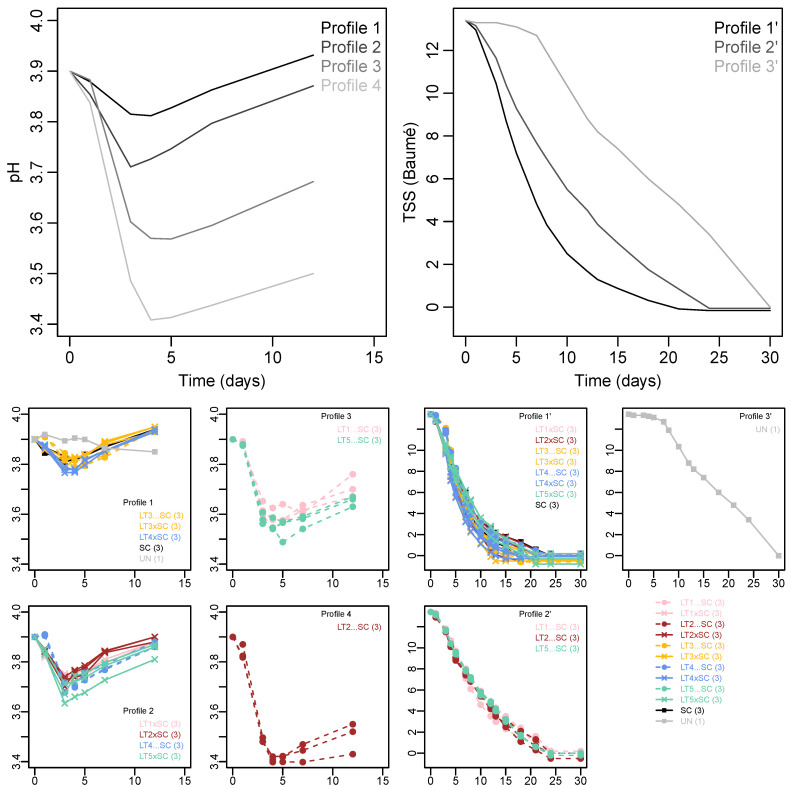
K-means clustering of acidification and fermentation kinetics in Viognier. Four and three profiles were resolved, respectively. The upper panels show the mean values of acidification kinetics (**left**) and fermentation kinetics profiles (**right**). The corresponding yeast treatments (with number of replicates in brackets) are indicated below the K-mean profiles. The yeast treatments include the *Saccharomyces cerevisiae* monoculture (SC), five *Lachancea thermotolerans* strains (LT1-LT5) in co-inoculations (xSC) or sequential inoculations (…SC) with *S. cerevisiae*, and un-inoculated treatment (UN).

**Figure 2 jof-08-00474-f002:**
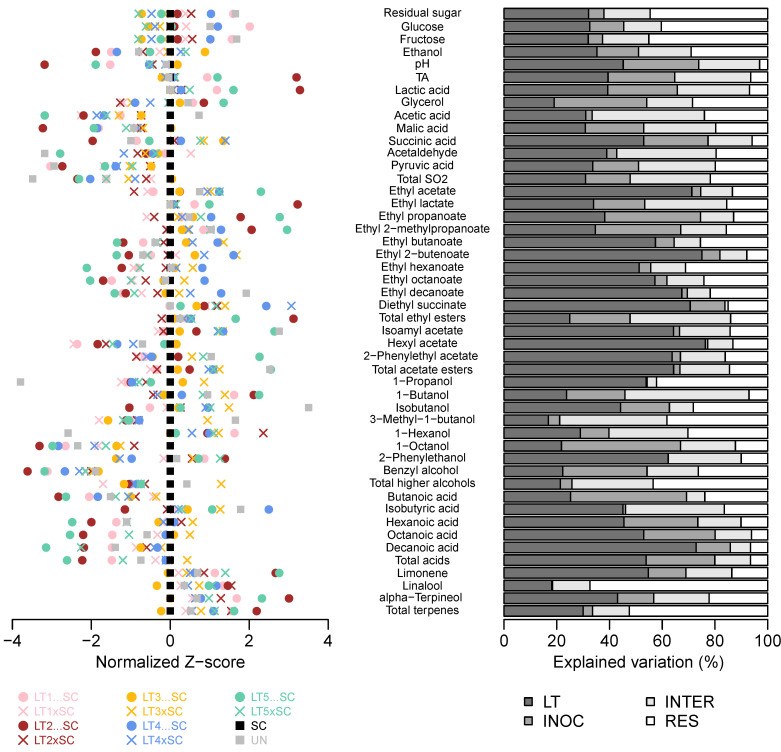
Variation in chemical composition of the experimental Viognier wines. Normalised Z-scores centered to SC wine (**left**). Percentages of variation in LT treatments explained by the LT strain (LT), inoculation modality (i.e., co-inoculation vs. sequential inoculation; INOC), their interaction (INTER) and residual (RES) as determined by 2-way ANOVA (**right**). The abbreviations and colour-coding of yeast treatments correspond to those in Figure 1.

**Figure 3 jof-08-00474-f003:**
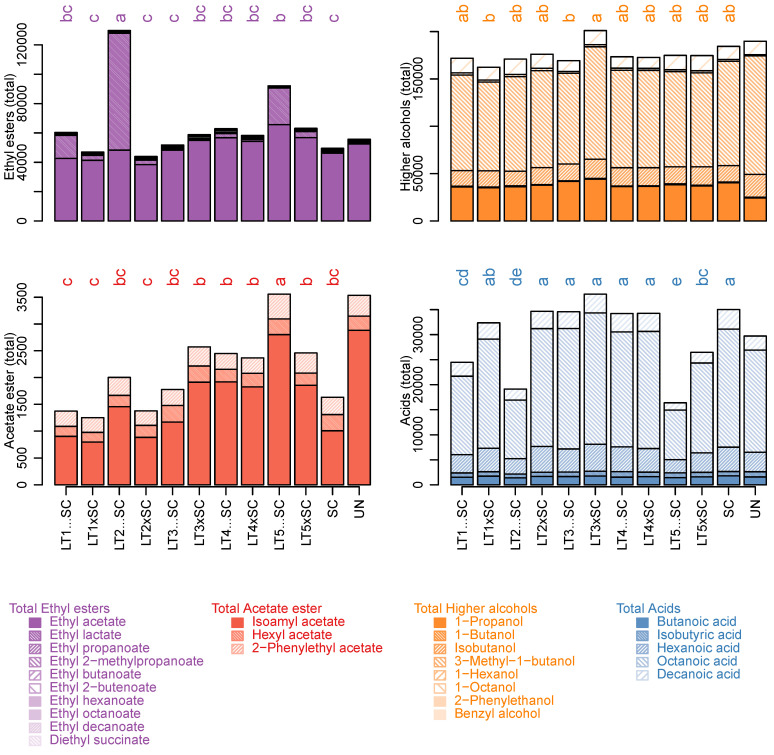
Sum of ethyl esters, acetate esters, higher alcohols and acids (µg/L) in experimental Viognier wines with contributions of individual compounds. The values represent means of triplicates and different letters represent significant differences (ANOVA; Tukey’s post-hoc α = 5%). The abbreviations of yeast treatments correspond to those in Figure 1.

**Figure 4 jof-08-00474-f004:**
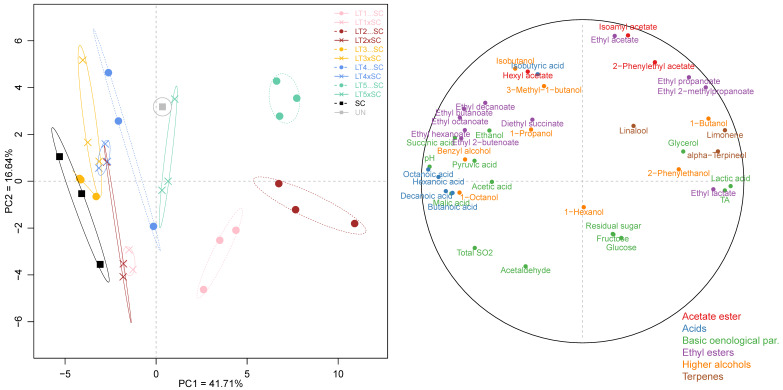
Principal component analysis of 43 chemical parameters in the experimental Viognier wines: yeast treatments (**left**) and correlation circle (**right**). The abbreviations and colour-coding of yeast treatments correspond to those in Figure 3.

**Figure 5 jof-08-00474-f005:**
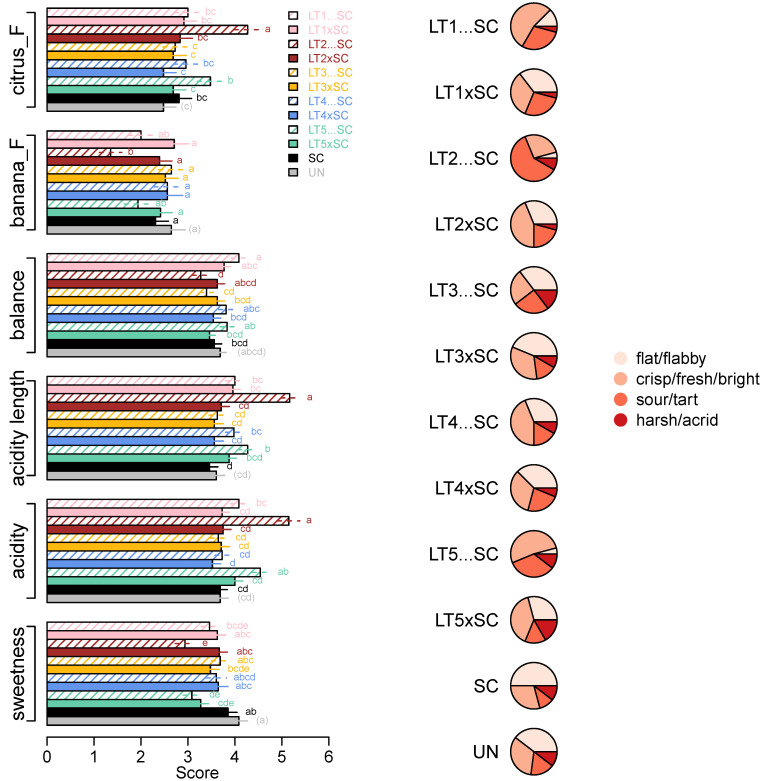
Intensity scores (means and standard errors) of sensory parameters significantly (ANOVA; Tukey’s post-hoc α = 5%; different letters represent significant differences) affected by yeast treatments (**left**) and acidity profiles of wines built with frequencies of four acidity descriptors (**right**; Appendix A).

**Table 1 jof-08-00474-t001:** Main oenological parameters of Viognier wines fermented with 12 yeast treatments. Values are the mean of winemaking triplicates and standard errors, and letters within the same row denote significance groups (ANOVA; Tukey’s post-hoc α = 5%). The yeast treatments include the *Saccharomyces cerevisiae* monoculture (SC), five *Lachancea thermotolerans* strains (LT1-LT5) in co-inoculations (xSC) or sequential inoculations (…SC) with *S. cerevisiae*, and an un-inoculated treatment (UN). The un-replicated treatment UN was excluded from the statistical analysis.

Parameters	Yeast Treatment											
	SC	LT1xSC	LT1…SC	LT2xSC	LT2…SC	LT3xSC	LT3…SC	LT4xSC	LT4…SC	LT5xSC	LT5…SC	UN
Glucose (g/L)	0.2 ± 0.0 ab	0.2 ± 0.0 ab	0.3 ± 0.1 a	0.2 ± 0.0 ab	0.2 ± 0.0 ab	0.2 ± 0.0 ab	0.1 ± 0.0 b	0.2 ± 0.0 b	0.3 ± 0.1 ab	0.1 ± 0.0 b	0.2 ± 0.0 ab	0.3
Fructose (g/L)	2.6 ± 1.0 a	2.6 ± 0.1 a	3.8 ± 1.4 a	2.7 ± 0.9 a	2.4 ± 0.5 a	2.0 ± 0.7 a	1.4 ± 0.5 a	1.7 ± 0.3 a	3.2 ± 1.5 a	1.3 ± 1.0 a	1.9 ± 0.3 a	3.9
Residual sugar (g/L)	2.8 ± 1.0 ab	2.8 ± 0.1 ab	4.1 ± 1.5 a	2.9 ± 0.9 ab	2.8 ± 0.5 ab	2.2 ± 0.7 ab	1.5 ± 0.5 ab	1.9 ± 0.4 ab	3.5 ± 1.6 ab	1.4 ± 1.0 b	2.1 ± 0.3 ab	4.2
Ethanol (% *v*/*v*)	14.8 ± 0.2 abc	14.7 ± 0.1 abc	14.5 ± 0.2 bc	14.7 ± 0.1abc	14.5 ± 0.1 c	14.8 ± 0.2 abc	14.9 ± 0.1 a	14.8 ± 0.0 ab	14.5 ± 0.1 bc	14.8 ± 0.1 ab	14.7 ± 0.1 abc	14.6
pH	4.14 ± 0.01 ab	4.05 ± 0.02 b	3.89 ± 0.07 c	4.06 ± 0.02 b	3.62 ± 0.07 d	4.15 ± 0.01 ab	4.17 ± 0.01 a	4.12 ± 0.01 ab	4.05 ± 0.02 b	4.05 ± 0.03 b	3.83 ± 0.03 c	4.12
TA (g/L)	4.6 ± 0.2 d	4.7 ± 0.1 cd	5.8 ± 0.4 bc	4.7 ± 0.1 cd	8.8 ± 1.1 a	4.4 ± 0.1 d	4.3 ± 0.1 d	4.5 ± 0.1 d	4.7 ± 0.1 cd	4.8 ± 0.2 cd	6.2 ± 0.5 b	4.55
Lactic acid (g/L)	0.1 ± 0.0 d	0.6 ± 0.1 cd	1.9 ± 0.5 bc	0.5 ± 0.1 cd	5.2 ± 1.4 a	0.1 ± 0.0 d	0.1 ± 0.0 d	0.2 ± 0.0 d	0.5 ± 0.1 cd	0.9 ± 0.3 cd	2.6 ± 0.5 b	0.10
Glycerol (g/L)	5.6 ± 0.1 abc	5.4 ± 0.0 c	5.7 ± 0.1 abc	5.4 ± 0.1 c	5.8 ± 0.1 ab	5.5 ± 0.1 bc	5.7 ± 0.1 abc	5.5 ± 0.1 abc	5.5 ± 0.2 bc	5.6 ± 0.1 abc	5.8 ± 0.1 a	5.69
Acetic acid (g/L)	0.45 ± 0.0 a	0.3 ± 0.0 cd	0.4 ± 0.0 abc	0.4 ± 0.0 ab	0.3 ± 0.0 cd	0.4 ± 0.1 abc	0.4 ± 0.0 ab	0.4 ± 0.0 bcd	0.3 ± 0.0 bc	0.4 ± 0.0 bc	0.3 ± 0.0 d	0.5
Malic acid (g/L)	3.0 ± 0.0 a	2.9 ± 0.0 ab	2.8 ± 0.0 bc	2.9 ± 0.0 ab	2.6 ± 0.1 c	2.9 ± 0.0 ab	3.0 ± 0.1 a	2.9 ± 0.0 ab	2.8 ± 0.1 bc	2.9 ± 0.1 ab	2.8 ± 0.0 bc	2.9
Succinic acid (g/L)	2.4 ± 0.2 bcd	2.5 ± 0.0 bc	2.1 ± 0.0 d	2.4 ± 0.1 bc	1.7 ± 0.1 e	2.7 ± 0.2 ab	2.9 ± 0.0 a	2.9 ± 0.0 a	2.5 ± 0.1 bc	2.7 ± 0.0 ab	2.2 ± 0.2 cd	2.0
Acetaldehyde (mg/L)	69 ± 4 ab	77 ± 10 a	67 ± 2 ab	56 ± 6 ab	59 ± 11 ab	59 ± 4 ab	64 ± 3 ab	51 ± 17 b	62 ± 3 ab	63 ± 1 ab	26 ± 1 c	20
Pyruvic acid (mg/L)	73 ± 3 a	54 ± 9 bcd	39 ± 4 d	67 ± 4 ab	42 ± 1 cd	61 ± 3 ab	67 ± 1 ab	58 ± 1 abc	57 ± 1 abc	62 ± 6 ab	54 ± 12 bcd	40
Total SO₂ (mg/L)	102 ± 3 a	95 ± 1 ab	81 ± 6 bcd	94 ± 9 ab	71 ± 2 d	88 ± 4 abcd	102 ± 7 a	90 ± 1 abc	75 ± 4 cd	81 ± 15 bcd	71 ± 2 d	56

**Table 2 jof-08-00474-t002:** Volatile composition of Viognier wines fermented with 12 yeast treatments. Volatile compounds in italics were detected below their sensory threshold in all wines. Compounds in italics and bold were in some wines below, and in others above, their sensory threshold (Appendix A). Values (µg/L) are the mean of winemaking triplicates with standard errors, and different letters within the same row represent significant differences (ANOVA; Tukey’s post-hoc α = 5%). The yeast treatments include the *Saccharomyces cerevisiae* monoculture (SC), five *Lachancea thermotolerans* strains (LT1-LT5) in co-inoculations (xSC) or sequential inoculations (…SC) with *S. cerevisiae*, and an un-inoculated treatment (UN). The un-replicated treatment UN was excluded from the statistical analysis.

Compound (µg/L)	Yeast Treatment											
	SC	LT1xSC	LT1…SC	LT2xSC	LT2…SC	LT3xSC	LT3…SC	LT4xSC	LT4…SC	LT5xSC	LT5…SC	UN
Ethyl acetate	46,234 ± 6244 bcd	41,344 ± 1261 d	42,619 ± 3249 cd	38,471 ± 882 d	48,287 ± 2939 bcd	54,898 ± 4916 ab	48,203 ± 1290 bcd	54,210 ± 1136 abc	56,729 ± 7592 ab	56,787 ± 5952 ab	65,667 ± 2677 a	52,518
*Ethyl lactate*	*649 ± 31 b*	*3401 ± 1149 b*	*15,716 ± 6545 b*	*3020 ± 1096 b*	*79,647 ± 37,474 a*	*809 ± 97 b*	*851 ± 55 b*	*1197 ± 39 b*	*2846 ± 461 b*	*4183 ± 502 b*	*24,946 ± 6322 b*	*522*
*Ethyl propanoate*	*78 ± 3 cde*	*74 ± 2 e*	*80 ± 1 cde*	*75 ± 4 de*	*92 ± 1 ab*	*80 ± 4 cde*	*83 ± 3 bcde*	*80 ± 1 cde*	*86 ± 6 bc*	*84 ± 5 bcd*	*100 ± 2 a*	*82*
** *Ethyl 2-methylpropanoate* **	** *14 ± 1 c* **	**14 ± 0 c**	** *15 ± 1 bc* **	**14 ± 1 c**	** *17 ± 1 ab* **	** *15 ± 1 c* **	**15 ± 0 c**	** *15 ± 0 c* **	** *16 ± 1 bc* **	** *16 ± 1 bc* **	** *19 ± 1 a* **	** *16* **
Ethyl butanoate	286 ± 48 abc	270 ± 8 abc	253 ± 20 bc	277 ± 41 abc	229 ± 44 c	350 ± 32 a	305 ± 7 abc	313 ± 13 abc	343 ± 55 ab	270 ± 26 abc	221 ± 5 c	267
Ethyl 2-butenoate	199 ± 23 cd	212 ± 12 cd	154 ± 33 def	194 ± 42 cde	103 ± 21 ef	354 ± 36 a	257 ± 8 bc	278 ± 18 abc	347 ± 74 ab	147 ± 9 def	75 ± 11 f	135
Ethyl hexanoate	961 ± 124 ab	756 ± 145 abc	779 ± 101 abc	936 ± 142 abc	688 ± 211 bc	1088 ± 112 ab	986 ± 96 ab	994 ± 51 ab	1141 ± 331 a	790 ± 109 abc	492 ± 33 c	965
Ethyl octanoate	863 ± 137 abc	670 ± 151 bcd	563 ± 58 cd	737 ± 42 abcd	519 ± 98 cd	935 ± 61 ab	828 ± 21 abc	836 ± 17 abc	1039 ± 318 a	661 ± 87 bcd	453 ± 27 d	838
** *Ethyl decanoate* **	** *212 ± 62 abc* **	** *165 ± 10 bc* **	** *133 ± 19 c* **	** *191 ± 35 abc* **	** *138 ± 24 bc* **	** *226 ± 20 abc* **	** *204 ± 14 abc* **	** *245 ± 11 ab* **	** *295 ± 86 a* **	** *153 ± 25 bc* **	** *120 ± 28 c* **	** *337* **
*Diethyl succinate*	*15 ± 3 d*	*30 ± 6 cd*	*15 ± 7 d*	*35 ± 15 bcd*	*30 ± 5 cd*	*39 ± 5 bc*	*27 ± 2 cd*	*67 ± 2 a*	*56 ± 12 ab*	*35 ± 6 bcd*	*20 ± 5 cd*	*15*
Ʃ Ethyl esters	49,511 ± 6619 c	46,935 ± 1881 c	60,327 ± 9869 bc	43,951 ± 2226 c	129,751 ± 34,148 a	58,792 ± 5211 bc	51,758 ± 1358 c	58,235 ± 1258 bc	62,899 ± 7980 bc	63,124 ± 6678 bc	92,112 ± 7272 b	55,695
Isoamyl acetate	1006 ± 201 cd	799 ± 107 d	905 ± 319 d	887 ± 320 d	1457 ± 82 bcd	1913 ± 246 b	1169 ± 66 bcd	1826 ± 285 bc	1919 ± 621 b	1856 ± 327 b	2802 ± 173 a	2881
*Hexyl acetate*	*302 ± 36 ab*	*179 ± 8 e*	*183 ± 21 e*	*220 ± 19 de*	*209 ± 14 de*	*303 ± 11 a*	*310 ± 39 a*	*254 ± 11 abcd*	*234 ± 34 bcde*	*228 ± 21 cde*	*294 ± 17 abc*	266
*2-Phenylethyl acetate*	*324 ± 32 bc*	*274 ± 12 c*	*285 ± 30 c*	*272 ± 28 c*	*336 ± 29 bc*	*357 ± 50 bc*	*297 ± 4 bc*	*287 ± 11 bc*	*296 ± 34 bc*	*375 ± 30 ab*	*462 ± 45 a*	387
Ʃ Acetate esters	1632 ± 254 bc	1251 ± 125 c	1373 ± 364 c	1378 ± 367 c	2003 ± 107 bc	2573 ± 302 b	1776 ± 101 bc	2368 ± 304 b	2448 ± 689 b	2459 ± 376 b	3558 ± 231 a	3535
1-Propanol	40,439 ± 2680 ab	35,153 ± 206 b	35,865 ± 1798 ab	37,804 ± 3774 ab	36,181 ± 1529 ab	44,086 ± 5844 a	41,799 ± 2323 ab	36,554 ± 522 ab	36,302 ± 2030 ab	37,061 ± 2909 ab	38,340 ± 2824 ab	24,268
*1-Butanol*	*667 ± 35 cd*	*643 ± 32 d*	*927 ± 66 ab*	*598 ± 36 d*	*1007 ± 28 a*	*889 ± 97 ab*	*640 ± 27 d*	*615 ± 41 d*	*714 ± 77 cd*	*799 ± 45 bc*	*1029 ± 38 a*	*819*
*Isobutanol*	*17,414 ± 1078 abc*	*17,252 ± 145 abc*	*16,431 ± 287 bc*	*17,909 ± 1539 abc*	*15,441 ± 487 c*	*20,252 ± 1939 a*	*17,807 ± 394 abc*	*19,127 ± 91 ab*	*19,241 ± 1635 ab*	*19,347 ± 1408 ab*	*17,912 ± 281 abc*	*24,093*
3-Methyl-1-butanol	110,102 ± 6608 ab	93,603 ± 1889 b	100,894 ± 4792 ab	102,420 ± 7067 ab	99,868 ± 1787 b	118,740 ± 10,969 a	95,691 ± 1478 b	102,791 ± 3739 ab	102,925 ± 7158 ab	99,172 ± 10406 b	100,527 ± 1325 b	125,227
*1-Hexanol*	*1959 ± 63 c*	*2224 ± 8 abc*	*2374 ± 15 ab*	*2560 ± 268 a*	*2199 ± 23 abc*	*2271 ± 218 abc*	*1959 ± 32 c*	*2099 ± 47 bc*	*2211 ± 95 abc*	*2240 ± 212 abc*	*1995 ± 43 c*	*1300*
1-Octanol	11 ± 1 a	6 ± 1 bcd	3 ± 1 cde	9 ± 1 ab	1 ± 1 e	8 ± 2 b	7 ± 0 b	6 ± 0 bcd	3 ± 1 de	6 ± 1 bc	2 ± 1 e	4
** *2-Phenylethanol* **	** *13,766 ± 1003 bc* **	** *13,470 ± 385 bcd* **	** *15,306 ± 501 ab* **	** *14,747 ± 757 ab* **	** *16,232 ± 370 a* **	** *14,809 ± 1371 ab* **	** *11,327 ± 505 d* **	** *11,484 ± 227 d* **	** *12,027 ± 445 cd* **	** *16,016 ± 1236 a* **	** *15,049 ± 350 ab* **	** *14,058* **
*Benzyl alcohol*	*61 ± 5 a*	*50 ± 0 b*	*45 ± 0 bc*	*51 ± 3 b*	*43 ± 1 c*	*51 ± 4 b*	*52 ± 1 b*	*49 ± 1 bc*	*47 ± 2 bc*	*50 ± 4 bc*	*45 ± 2 bc*	*51*
Ʃ Higher alcohols	184,420 ± 11,304 ab	162,401 ± 2360 b	171,845 ± 6827 ab	176,096 ± 13,152 ab	170,972 ± 898 ab	201,107 ± 20,264 a	169,281 ± 2275 b	172,724 ± 4266 ab	173,469 ± 10,602 ab	174,691 ± 16,200 ab	174,899 ± 3271 ab	189,821
Butanoic acid	1791 ± 79 a	1731 ± 151 abc	1506 ± 26 cde	1669 ± 94 abc	1398 ± 93 e	1741 ± 84 ab	1652 ± 69 abcd	1643 ± 59 abcd	1536 ± 62 bcde	1599 ± 18 abcde	1424 ± 22 de	1580
*Isobutyric acid*	*868 ± 50 bc*	*888 ± 12 bc*	*873 ± 33 bc*	*860 ± 51 bc*	*760 ± 5 c*	*988 ± 84 ab*	*910 ± 21 b*	*888 ± 12 bc*	*1104 ± 45 a*	*895 ± 82 bc*	*968 ± 42 ab*	*1036*
Hexanoic acid	4878 ± 370 ab	4685 ± 235 abc	3654 ± 198 cde	5130 ± 569 a	3086 ± 733 de	5389 ± 327 a	4614 ± 69 abc	4724 ± 60 ab	4968 ± 68 a	3890 ± 415 bcd	2651 ± 156 e	3885
Octanoic acid	23,557 ± 1671 a	21,808 ± 435 ab	15,657 ± 1225 cd	23,543 ± 2967 a	11,675 ± 3038 de	26,229 ± 1964 a	24,061 ± 699 a	23,393 ± 495 a	22,937 ± 1432 a	17,940 ± 1199 bc	9880 ± 466 e	20,395
Decanoic acid	3935 ± 86 a	3253 ± 75 ab	2764 ± 267 bc	3460 ± 435 a	2200 ± 252 c	3752 ± 153 a	3341 ± 101 ab	3616 ± 90 a	3679 ± 127 a	2143 ± 221 c	1441 ± 396 d	2828
Ʃ Acids	35,028 ± 2166 a	32,365 ± 698 ab	24,454 ± 1653 cd	34,662 ± 4111 a	19,119 ± 4100 de	38,098 ± 2597 a	34,577 ± 851 a	34,264 ± 586 a	34,224 ± 1597 a	26,467 ± 1920 bc	16,365 ± 1002 e	29,724
*Limonene*	*4.0 ± 0.3 d*	*4.1 ± 0.1 cd*	*4.8 ± 0.2 cd*	*4.6 ± 0.6 cd*	*6 ± 0.2 ab*	*4.3 ± 0.5 cd*	*3.9 ± 0 d*	*4.3 ± 0.1 cd*	*4.4 ± 0.2 cd*	*5 ± 0.6 bc*	*6.1 ± 0.2 a*	*4.5*
*a-terpineol*	*41 ± 3 d*	*45 ± 3 cd*	*53 ± 6 abc*	*50 ± 8 bcd*	*63 ± 1 a*	*46 ± 4 bcd*	*41 ± 1 d*	*45 ± 1 cd*	*46 ± 2 bcd*	*48 ± 6 bcd*	*58 ± 3 ab*	*46*
Linalool	112 ± 12 a	116 ± 5 a	130 ± 10 a	133 ± 25 a	132 ± 8 a	122 ± 21 a	108 ± 2 a	118 ± 2 a	128 ± 10 a	127 ± 21 a	125 ± 2 a	117
Ʃ Terpenes	157 ± 15 a	165 ± 8 a	188 ± 16 a	188 ± 33 a	201 ± 8 a	172 ± 25 a	152 ± 3 a	168 ± 3 a	179 ± 10 a	179 ± 27 a	189 ± 5 a	167

## Data Availability

All relevant data are within the article and the Appendix A.

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
