# Peer review of "Impact of Lachancea thermotolerans on Chemical Composition and Sensory Profiles of Viognier Wines"

_jof, 2022, doi:10.3390/jof8050474_

Round 1

Reviewer 1 Report

The manuscript describes the use of Lachancea thermotolerans for the biological acidification of wine obtained from the Viognier variety cultivated in warm climate environments. Five strains of L. thermotolerans were compared in two different inoculation modes (co-inoculation and sequential) with Saccharomyces cerevisiae. All wines were analysed from a chemical-physical and sensory aspect. Volatile organic compounds were determined for each wine.
The general presentation is clear and concise. The experimental approach is fit for purpose, the materials and methods used to perform the different assays are adequately described, and the statistical treatment of the resulting data is appropriate. 
I have only a few minor revisions to suggest to the authors to make the manuscript suitable for publication in JoF:

Line 107: 2.1. Grapes and winemaking...In this paragraph the authors should specify the sampling plan.
Lines 114-115: How many glass demijohns were used for each treatment?
Line 146: The analysis of 29 volatile compounds...how do the authors in this section know how many volatile organic compounds will be determined? I would delete the number 29.
Line 248-250: After how many days has alcoholic fermentation been completed?
Table 1
1. The values in table 1 must be standardised. 
2. How many digits after the decimal point? One or two? Making the reported data homogeneous. 
3. Enter the letters of the significant differences between the trials for fructose.
4. Is it possible to report on the data of yeast assimilable nitrogen?
5. Why didn't the authors report free SO2 values?
Table 2: For linalool and terpenes, letters of significance between treatments are missing.
Lines 538-539. Delete sentence "This section...."

Author Response

Responses to individual comments are detailed below. In addition some other minor edits/corrections have been made. 

Line 107: 2.1. Grapes and winemaking...In this paragraph the authors should specify the sampling plan.
RESPONSE: Harvest criteria (Bé) added.

Lines 114-115: How many glass demijohns were used for each treatment?           

RESPONSE: Replication details provided on Ln 130.

Line 146: The analysis of 29 volatile compounds...how do the authors in this section know how many volatile organic compounds will be determined? I would delete the number 29.
RESPONSE: Done

Line 248-250: After how many days has alcoholic fermentation been completed?

RESPONSE: The varied according to the strain and inoculation modality. Details are provided in Fig 1. Averages are mentioned in the text – Ln 227-233.

Table 1: 
1. The values in table 1 must be standardised. 

RESPONSE: decimal places reduced/standardised as appropriate.

    How many digits after the decimal point? One or two? Making the reported data homogeneous. 

RESPONSE: decimal places reduced/standardised as appropriate for the analyte.

    Enter the letters of the significant differences between the trials for fructose.

RESPONSE: Done

    Is it possible to report on the data of yeast assimilable nitrogen?

RESPONSE: Unfortunately, this was not collected and cannot be done now.

    Why didn't the authors report free SO2values?

RESPONSE: Total SO2 seemed most relevant and happened to be the analysis that was conducted.

Table 2: For linalool and terpenes, letters of significance between treatments are missing.

RESPONSE: Done.

Lines 538-539. Delete sentence "This section...."

RESPONSE: Done.

Reviewer 2 Report

Thank you for a well written manuscript. There are a few minor adjustments needed.

Line 5: Should the ‘and’ not be moved to after author ‘Bely’?

Line 96: Remove '>' after [26].

Line 115: As degree Brix are used in many wine industries, it will be helpful to the reader if the approximate equivalent Brix value could be given in brackets after the Baume value.

Line 193: ‘break’ not ‘beak’.

Line 194: Replace ‘they’ with ‘assessors’.

Line 195: space between ‘carry’ and ‘over’.

Line 218: Data set is usually two words.

Line 333: Maybe insert the aroma in brackets after isoamyl acetate (similar to what is done in Line 352). Also for the other compounds, even if below sensory threshold, as they can collectively still play a role.

Line 373: Insert the word ‘only’ before 41,71%. Collectively the PCA only accounted 58.35% of the variance, which is on the low side.

Figure 4 legend. It should probably read ‘… correspond to those in Figure ‘3’ ?

Lines 427 – 428: Is this sentence referring to other published work? Then a reference is required. Or is it referring to this study, then the sentence needs to be rephrased.

Line 454: Correct placement of comma between ‘recently’ and ‘Sgouros’.

Line 538 – 539: This sentence should be removed.

Author Response

Responses to individual comments are detailed below. In addition some other minor edits/corrections have been made. 

Reviewer Comments:

Line 5: Should the ‘and’ not be moved to after author ‘Bely’?

RESPONSE: Done.

Line 96: Remove '>' after [26].

RESPONSE: Done.

Line 115: As degree Brix are used in many wine industries, it will be helpful to the reader if the approximate equivalent Brix value could be given in brackets after the Baume value.

RESPONSE: Done – Ln 120

Line 193: ‘break’ not ‘beak’.

RESPONSE: Done.

Line 194: Replace ‘they’ with ‘assessors’.

RESPONSE: Done.

Line 195: space between ‘carry’ and ‘over’.

RESPONSE: Done.

Line 218: Data set is usually two words.

RESPONSE: Done.

Line 333: Maybe insert the aroma in brackets after isoamyl acetate (similar to what is done in Line 352). Also for the other compounds, even if below sensory threshold, as they can collectively still play a role.

RESPONSE: Done for many compounds in this section.

Line 373: Insert the word ‘only’ before 41,71%. Collectively the PCA only accounted 58.35% of the variance, which is on the low side.

RESPONSE: Done.

Figure 4 legend. It should probably read ‘… correspond to those in Figure ‘3’ ?

RESPONSE: Yes, added.

Lines 427 – 428: Is this sentence referring to other published work? Then a reference is required. Or is it referring to this study, then the sentence needs to be rephrased.

RESPONSE: Rephrased to refer to this study.

Line 454: Correct placement of comma between ‘recently’ and ‘Sgouros’.

RESPONSE: Space added.

Line 538 – 539: This sentence should be removed.

RESPONSE: Done.